# A Thermopile Sensor Revealed That the Average Peripheral Wrist Skin Temperature of Patients with Major Depressive Disorder at 09:00 Is 2.9 °C Lower than That of Healthy People

**DOI:** 10.3390/s25051582

**Published:** 2025-03-05

**Authors:** Keisuke Watanabe, Shohei Sato, Yusuke Obara, Nobutoshi Kariya, Toshikazu Shinba, Takemi Matsui

**Affiliations:** 1Department of Electrical Engineering and Computer Science, Graduate School of Systems Design, Tokyo Metropolitan University, Tokyo 191-0065, Japan; watanabe-keisuke2@ed.tmu.ac.jp (K.W.); sato.shohei@tmu.ac.jp (S.S.); 2Maynds Tower Mental Clinic, Tokyo 151-0053, Japan; obara@mmc-mcc.com (Y.O.);; 3Department of Psychiatry, Shizuoka Saiseikai General Hospital, Shizuoka 422-8527, Japan; t156591@siz.saiseikai.or.jp

**Keywords:** diurnal variation, heart rate, major depressive disorder, morning sympathetic surge, peripheral wrist skin temperature

## Abstract

Many patients with major depressive disorder (MDD) feel worse in the morning than in the evening. To clarify the differences in morning physiological characteristics between patients with MDD and healthy participants, a wearable device that measures peripheral wrist skin temperature and heart rate (HR) was adopted. The device incorporates a thermopile sensor to measure peripheral wrist skin temperature using infrared radiation emitted from the skin surface. In total, 30 patients diagnosed with MDD and 24 healthy individuals were recruited. From 00:00 to 12:00, participants wore a wrist-worn device on their non-dominant hand. It was discovered that, at 09:00, the average peripheral wrist skin temperature of patients with MDD was significantly lower (by 0.1% [2.9 °C]) than that of healthy individuals. The dramatic decrease in morning (09:00) peripheral wrist skin temperature in patients with MDD can be attributed to their morning sympathetic surge and peripheral vascular contraction. The average HR of patients with MDD was significantly higher (by 1% [17 beats/min]) than that of healthy controls. Regression analysis, including peripheral wrist skin temperature and HR at 09:00, showed 83.3% sensitivity and a negative predictive value of 76.2%. The potential impact of these results appears promising for future preliminary morning MDD screening.

## 1. Introduction

Diurnal variation in mood is one of the symptoms of major depressive disorder (MDD) [1]. Many patients with MDD feel worse in the morning than in the evening [2]. Balter et al. investigated diurnal changes in psychiatric symptoms and behaviors through questionnaires, revealing worse psychiatric and behavior scores in the morning for patients with MDD [3]. The relationship between diurnal variation and biochemical characteristics has attracted attention. Gold et al. found that plasma levels of norepinephrine, a neurotransmitter in the sympathetic nervous system, in patients with MDD peaked in the morning [4].

To clarify the differences in morning physiological characteristics between patients with MDD and healthy participants, physiological parameters, i.e., peripheral wrist skin temperature and heart rate (HR), were analyzed. This report examines the differences in morning physiological characteristics between patients with MDD and healthy participants. In previous studies, specific physiological characteristics of patients with MDD were used to facilitate MDD screening [5,6]. Accordingly, screening for patients with MDD was conducted based on the differences in morning physiological characteristics between patients with MDD and healthy individuals.

## 2. Materials and Methods

A wrist-worn device (E4 wristband, Empatica Inc., Cambridge, MA, USA) was used to measure peripheral (outside part of the wrist) skin temperature and HR. The device incorporates a thermopile sensor that measures infrared radiation emitted from the peripheral skin surface at a sampling rate of 4 Hz. The thermopile sensor can measure skin temperature with a resolution of 0.02 °C over a range from −40 °C to 115 °C, with an accuracy of ±0.2 °C within the range of 36 °C to 39 °C. Notably, it enables accurate temperature measurements despite a small gap between the skin surface and the device. The wrist-worn device also incorporates a photoplethysmography sensor for HR measurement. HR was calculated from the inter-beat interval provided by the device. The series of HRs were linearly interpolated and resampled at 1 Hz to account for the uneven spacing of inter-beat intervals. A total of 30 patients with MDD (age, 39 ± 9 years; 17 females, 13 males), diagnosed on the International Statistical Classification of Diseases and Related Health Problems at Maynds Tower Mental Clinic (Tokyo, Japan), were recruited. In total, 24 healthy individuals (age, 30 ± 13 years; 14 females, 10 males) were recruited at Tokyo Metropolitan University (Tokyo, Japan). From 00:00 to 12:00, participants wore the device on the outside of their non-dominant wrist to minimize motion artifacts. Peripheral wrist skin temperature and HR were compared between the MDD group and the healthy group using the Mann–Whitney U test every hour, implemented in Python 3.9.12 with SciPy 1.9.3. Logistic regression analysis with scikit-learn 1.0.2, performed using peripheral wrist skin temperature and HR as explanatory variables, identified the time when the difference in wrist peripheral skin temperature between the two groups was greatest. Classification was performed using only the features at that time point to distinguish between MDD patients and healthy controls.

## 3. Results and Discussion

At 09:00, the peripheral wrist skin temperature of patients with MDD was found to be significantly lower, by 2.9 °C (equivalent to 0.1%), than that of healthy individuals according to the thermopile sensor of the wrist-worn device (30.9 ± 3.5 °C and 33.8 ± 0.9 °C, respectively, *p* < 0.001), as shown in Figure 1. The peripheral wrist skin temperature difference between the MDD and the healthy groups reached its maximum at 09:00 (green dotted line in Figure 1). A previous study found that the core body temperature of patients with MDD is higher than that of healthy controls at 09:00 [7]. It is worth mentioning that the peripheral wrist skin temperature and core body temperature of patients with MDD exhibit opposite trends.

The dramatic decrease in peripheral wrist skin temperature at 09:00 in patients with MDD can be attributed to morning sympathetic surge-induced peripheral vascular contraction [4,8,9]. Morning sympathetic surge may occur around 09:00 because the HR of patients with MDD at that time is significantly higher than that of healthy individuals (88 ± 20 and 72 ± 14 beats/min, respectively, *p* < 0.01; red dotted line in Figure 1). HR is correlated with sympathetic nervous activity, as shown by its decrease after pharmacological sympathetic blockade with β-blockers [10], and a morning surge in sympathetic nervous activity is a well-known phenomenon [8]. Moreover, peripheral skin temperature can be influenced by peripheral vascular contraction. Herborn et al. revealed that peripheral temperature changes in proportion to quantitative stressor intensity, which affects sympathetic activation in animal experiments [11]. In this study, the decrease in peripheral wrist skin temperature at 09:00 in patients with MDD can be attributed to a morning increase in plasma norepinephrine, which induces peripheral vascular contraction [4,12,13].

The following logistic regression equation was derived from the peripheral wrist skin temperature (°C) and HR (beats/min) at 09:00:*Logit score* = −0.18 ⋅ *peripheral wrist skin temperature* + 0.07 ⋅ *HR* + 0.47
where a logit score ≥ 0 corresponds to suspected MDD and a score < 0 corresponds to healthy.

Using only peripheral wrist skin temperature and HR, the proposed algorithm achieved 83.3% sensitivity and a negative predictive value of 76.2%. These metrics are valuable in the first stage of MDD screening. Although the screening accuracy is not sufficiently high, the proposed method has the potential to facilitate psychiatric consultations with potential patients. Sun et al. and Kuang et al. successfully distinguished patients with MDD from healthy controls within approximately five to ten minutes by measuring changes in heart rate variability (HRV) induced by a random number generation and Ewing test, respectively [5,14]. However, that protocol is limited to hospital settings. As an alternative method applicable outside of hospitals, Sato et al. employed a wrist-worn device and a sleep relaxation intervention instead of stress tasks to achieve high accuracy in MDD screening [6]. The proposed method, based only on peripheral wrist skin temperature and HR at 09:00, appears promising for future preliminary screening of MDD within a few seconds outside of hospitals.

This study has several limitations. First, the sample size was relatively small, which may limit the generalizability of the findings. Second, as the study population included patients, their activities were not restricted. Consequently, the measurement environment could not be fully standardized.

## 4. Conclusions

The peripheral wrist skin temperature of patients with MDD at 09:00, detected by a thermopile sensor in a wrist-worn device, was found to be significantly lower than that of healthy individuals by 2.9 °C (30.9 ± 3.5° and 33.8 ± 0.9°, respectively, *p* < 0.001). The dramatic decrease in peripheral wrist skin temperature at 09:00 in patients with MDD can be attributed to morning sympathetic surge-induced peripheral vascular contraction. The morning physiological characteristics could be applied to screen for patients with MDD. Logistic regression analysis, including peripheral wrist skin temperature and HR at 09:00, showed 83.3% sensitivity and a negative predictive value of 76.2%.

## 5. Patents

This screening algorithm is patent-pending in Japan (application number: 2024-109147).

## Figures and Tables

**Figure 1 sensors-25-01582-f001:**
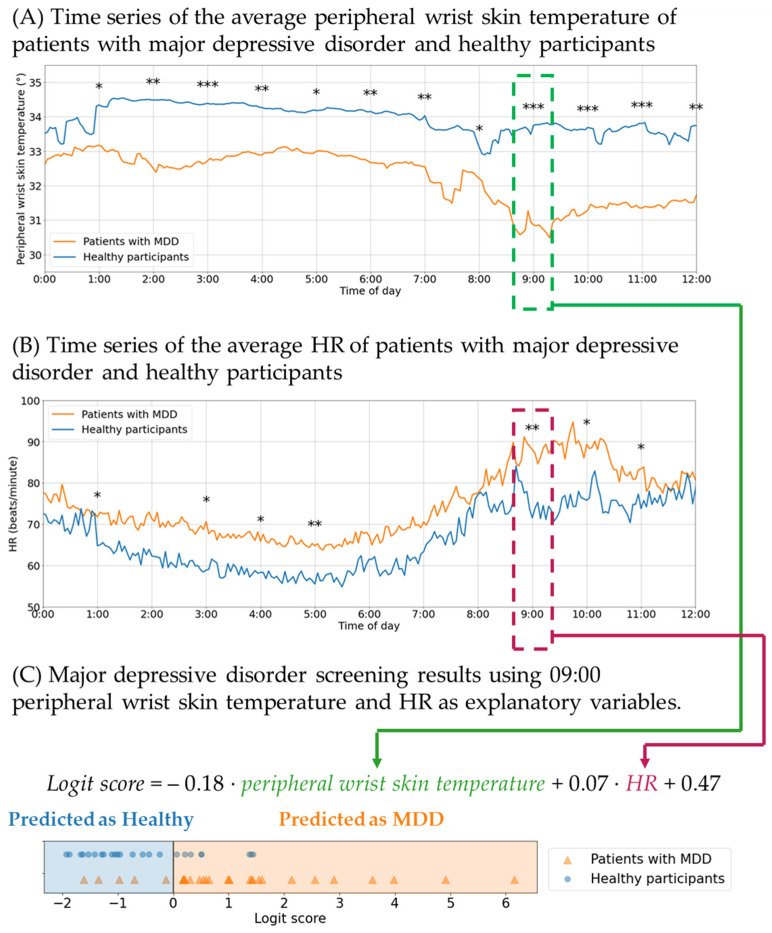
At 09:00, the peripheral wrist skin temperature (green dots) of patients with major depressive disorder (MDD) was significantly lower than that of healthy participants by 0.1% (30.9 ± 3.5 °C and 33.8 ± 0.9 °C, respectively, *p* < 0.001). At the same time point, the heart rate (HR; red dots) of patients with MDD was significantly higher than that of healthy individuals by approximately 1% (88 ± 20 and 72 ± 14 beats/min, respectively; *p* < 0.01). (**A**) Time series of the average peripheral wrist skin temperature from 00:00 to 12:00 (orange line, patients with MDD; blue line, healthy participants). (**B**) Time series of the average HR from 00:00 to 12:00 (orange line, patients with MDD; blue line, healthy participants). Mann–Whitney U test-derived significant differences are indicated by asterisks (* *p* < 0.05, ** *p* < 0.01, *** *p* < 0.001). (**C**) Using peripheral wrist skin temperature and HR at 09:00 as explanatory variables, logistic regression analysis achieved 83.3% sensitivity and a negative predictive value of 76.2% (logit score ≥ 0, suspected MDD; logit score < 0, healthy).

## Data Availability

The data are not publicly available due to privacy and ethical restrictions.

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
