# Peer review of "A Thermopile Sensor Revealed That the Average Peripheral Wrist Skin Temperature of Patients with Major Depressive Disorder at 09:00 Is 2.9 °C Lower than That of Healthy People"

_sensors, 2025, doi:10.3390/s25051582_

Round 1

Reviewer 1 Report

Comments and Suggestions for Authors

The manuscript aims to clarify the differences in morning physiological characteristics (peripheral wrist skin temperature and heart rate) between patients with major depressive disorder and healthy participants.
Results of regression analysis including peripheral wrist skin temperature and HR at 09:00 showed 83.3% sensitivity and a negative predictive value of 76.2%. 
I find the topic interesting and being worth of investigation.
The document is well structure, written, the background and references are sufficient and up to date, the methodology is well explained and the results and discussion adequately presented and the conclusions are supported by them.
Although, I have the following comments/suggestions:
- Abstract should be better organized: problem, motivation, aim, methodology, main results, further impact of those results.
- English language should be reviewed by a native speaker.
- Keywords should be alphabetically ordered.
- I strongly suggest authors from refraining using personal pronouns such as "we" and "our" throughout the text and I encourage them to write it in an impersonal form of writing.
- At materials and methods the thermopile sensor specifications must be described.

Comments on the Quality of English Language
  • English language should be reviewed by a native speaker.
  • I strongly suggest authors from refraining using personal pronouns such as "we" and "our" throughout the text and I encourage them to write it in an impersonal form of writing.

Author Response

We sincerely appreciate your valuable comments and suggestions on our manuscript titled “A Thermopile Sensor Revealed that the Average Peripheral Wrist Skin Temperature of Patients with Major Depressive Disorder at 09:00 is 2.9°C Lower than that of Healthy People” (Manuscript ID: sensors-3488899). We thoroughly considered each of your comments and revised our manuscript accordingly. Below, we provide a detailed response to each comment.

Comments 1:

Abstract should be better organized: problem, motivation, aim, methodology, main results, further impact of those results.

Response 1:

We sincerely apologize for the poor structure and revised the abstract according to your advice as follows.

Abstract: Many patients with major depressive disorder (MDD) feel worse in the morning than in the evening. To clarify the differences in morning physiological characteristics between patients with MDD and healthy participants, a wearable device that measures peripheral wrist skin temperature and heart rate (HR) was adopted. The device incorporates a thermopile sensor to measure peripheral wrist skin temperature using infrared radiation emitted from the skin surface. 30 patients diagnosed with MDD and 24 healthy individuals were recruited. From 00:00 to 12:00, participants wore a wrist-worn device on the non-dominant hand. It was discovered that, at 09:00, the average peripheral wrist skin temperature of patients with MDD was significantly lower (by 0.1% [2.9°C]) than that of healthy individuals. The dramatic decrease in morning (09:00) peripheral wrist skin temperature in patients with MDD can be attributed to their morning sympathetic surge and peripheral vascular contraction. The average HR of patients with MDD was significantly higher (by 1% [17 beats/minute]) than that of healthy controls. Regression analysis, including peripheral wrist skin temperature and HR at 09:00, showed 83.3% sensitivity and a negative predictive value of 76.2%. The potential impact of these results appears promising for future preliminary morning MDD screening.

Comments 2:

English language should be reviewed by a native speaker.

Response 2:

Our manuscript was initially proofread by Michael Irvine, PhD, from Edanz. However, following the incorporation of corrections, some errors inadvertently occurred. Due to the limited timeframe for manuscript resubmission, we undertook a meticulous examination to ensure its accuracy.

Comments 3:

Keywords should be alphabetically ordered.

Response 3:

We sincerely apologize for the mistakes in our manuscript and revised the order of the keywords according to your advice as follows.

Keywords: diurnal variation; heart rate; major depressive disorder; morning sympathetic surge; peripheral wrist skin temperature

Comments 4:

I strongly suggest authors from refraining using personal pronouns such as "we" and "our" throughout the text and I encourage them to write it in an impersonal form of writing.

Response 4:

Thank you for the comment that further enhances our manuscript. We revised them into an impersonal form following your advice.

Comments 5:

At materials and methods the thermopile sensor specifications must be described.

Response 5:

With your advice, we added the specifications of thermopile sensor as follows on page 2, line 50 to line 52 in the Materials and Methods section.

“The thermopile sensor can measure skin temperature with a resolution of 0.02°C over a range from -40°C to 115°C, with an accuracy of ±0.2°C within the range of 36°C to 39°C.”

We sincerely hope that our revisions have addressed your concerns and improved the quality of our manuscript. Once again, we are grateful for your insightful comments and suggestions, which have greatly contributed to enhancing our work. Thank you for your time and consideration.

Reviewer 2 Report

Comments and Suggestions for Authors

The manuscript entitled entails an interesting and important discovery. The authors mention that patients with major depressive disorder have a lower peripheral wrist skin temperature of 2.9°C and significantly higher heart rate than healthy individuals at 9 am. For the benefit of the reader, some information requires further explanation. There are given below.

  1. The sample capacity is 30 patients (age, 39±9 years; 17 females, 13 males) diagnosed with major depressive disorder and 24 healthy individuals (age, 30±13 years; 14 females, 10 males). The relatively small sample capacity may affect the accuracy of the conclusion. Please expand the sample capacity.
  2. In addition, there are differences in age distribution (with an average of 39 years for patients and 30 years for healthy individuals). Age is a potential influencing factor that may affect the accuracy of conclusions.
  3. I prefer to know if all patients and healthy individuals were tested under the same experimental conditions, such as room temperature.
  4. The authors mention that the peripheral wrist skin temperature and core body temperature of patients with MDD exhibit opposite trends. Please try to explain the physiological mechanism of this phenomenon.

Author Response

We sincerely appreciate your valuable comments and suggestions on our manuscript titled “A Thermopile Sensor Revealed that the Average Peripheral Wrist Skin Temperature of Patients with Major Depressive Disorder at 09:00 is 2.9°C Lower than that of Healthy People” (Manuscript ID: sensors-3488899). We thoroughly considered each of your comments and revised our manuscript accordingly. Below, we provide a detailed response to each comment.

Comments 1:

The sample capacity is 30 patients (age, 39±9 years; 17 females, 13 males) diagnosed with major depressive disorder and 24 healthy individuals (age, 30±13 years; 14 females, 10 males). The relatively small sample capacity may affect the accuracy of the conclusion. Please expand the sample capacity.

Response 1:

We appreciate your valuable comment. We acknowledge that the sample size in our study was relatively small, which may limit the generalizability of the findings. Before conducting the analysis, we performed a power analysis to estimate the adequate sample size. Considering that the accuracy of the adopted thermopile was ±0.2℃ within the range of 36℃ to 39℃, we aimed to detect a difference of 0.4℃. The minimum sample size was estimated to be 30 based on the power analysis with a power of 0.8 and a significance level of 0.05.

Nevertheless, we must admit that the sample size is relatively small. The decrease in peripheral wrist skin temperature observed at 09:00 in patients with MDD seems reasonable, as it can be attributed to a morning increase in plasma norepinephrine, which induces peripheral vascular contraction, leading to a decrease in peripheral wrist skin temperature.

Regrettably, we cannot expand the sample size by the deadline because building a trusting relationship with patients takes a long time. Therefore, we plan to increase the sample size in future studies.

Based on your comment, we have added the following description on page 3, from line 114 to line 115 in the Results and Discussion section.

“First, the sample size was relatively small, which may limit the generalizability of the findings.”

Comments 2:

In addition, there are differences in age distribution (with an average of 39 years for patients and 30 years for healthy individuals). Age is a potential influencing factor that may affect the accuracy of conclusions.

Response 2:

We appreciate your insight into our study. As pointed out, further investigation is needed to enable adjustment of the age distribution between patients with MDD and healthy volunteers. We anticipate that increasing the sample size will help reduce the mean age difference between the two groups, thereby enabling better adjustment of the age distribution.

Comments 3:

I prefer to know if all patients and healthy individuals were tested under the same experimental conditions, such as room temperature.

Response 3:

Unfortunately, due to the nature of our study, we were unable to control room temperature for each data collection. With this comment, we added the following description on page 3, line 115 to line 117 in the Results and Discussion section.

“Second, as the study population included patients, their activities were not restricted. Consequently, the measurement environment could not be fully standardized.”

Comments 4:

The authors mention that the peripheral wrist skin temperature and core body temperature of patients with MDD exhibit opposite trends. Please try to explain the physiological mechanism of this phenomenon.

Response 4:

The patients with MDD show higher sympathetic nervous activation than that of healthy volunteers. In peripheral vascular, sympathetic nervous activation promotes vascular contraction and peripheral temperature decrease. Consequently, this promotes concentration of blood in the center of the body. As a result, the core body temperature increases. We support this hypothesis, and our study findings do not contradict it. However, further research is needed to directly test this hypothesis.

We sincerely hope that our revisions have addressed your concerns and improved the quality of our manuscript. Once again, we are grateful for your insightful comments and suggestions, which have greatly contributed to enhancing our work. Thank you for your time and consideration.

Reviewer 3 Report

Comments and Suggestions for Authors

The authors found some unique results using a thermopile sensor in a wrist-worn device, that the peripheral wrist skin temperature of patients with symptoms of major depressive disorder (MDD) at 09:00 was significantly lower than that of healthy individuals by ~3°C (i.e., 30.9 ± 3.5° as compared to 33.8 ± 0.9°). This might be attributed to morning sympathetic surge-induced peripheral vascular contraction.

The results may be applied to screen patients with MDD, therefore are useful for clinic applications. However, the conclusion is not fully reliable, as the number of tested samples (people) is not large enough. 

Author Response

We sincerely appreciate your valuable comments and suggestions on our manuscript titled “A Thermopile Sensor Revealed that the Average Peripheral Wrist Skin Temperature of Patients with Major Depressive Disorder at 09:00 is 2.9°C Lower than that of Healthy People” (Manuscript ID: sensors-3488899). We thoroughly considered your comment and revised our manuscript accordingly. Below, we provide a detailed response to the comment.

Comment:

The results may be applied to screen patients with MDD, therefore are useful for clinic applications. However, the conclusion is not fully reliable, as the number of tested samples (people) is not large enough. 

Response:

We appreciate your valuable comment. We acknowledge that the sample size in our study was relatively small, which may limit the generalizability of the findings. Before conducting the analysis, we performed a power analysis to estimate the adequate sample size. Considering that the accuracy of the adopted thermopile was ±0.2℃ within the range of 36℃ to 39℃, we aimed to detect a difference of 0.4℃. The minimum sample size was estimated to be 30 based on the power analysis with a power of 0.8 and a significance level of 0.05.

Nevertheless, we must admit that the sample size is relatively small. The decrease in peripheral wrist skin temperature observed at 09:00 in patients with MDD seems reasonable, as it can be attributed to a morning increase in plasma norepinephrine, which induces peripheral vascular contraction, leading to a decrease in peripheral wrist skin temperature.

Regrettably, we cannot expand the sample size by the deadline because building a trusting relationship with patients takes a long time. Therefore, we plan to increase the sample size in future studies.

Based on your comment, we have added the following description from on page 3, line 114 to line 115 in the Results and Discussion section.

“First, the sample size was relatively small, which may limit the generalizability of the findings.”

We sincerely hope that our revisions have addressed your concern and improved the quality of our manuscript. Once again, we are grateful for your insightful comment, which have greatly contributed to enhancing our work. Thank you for your time and consideration.